# The Presence of a Virulent Clone of *Leptospira interrogans* Serovar Canicola in Confirmed Cases of Asymptomatic Dog Carriers in Mexico

**DOI:** 10.3390/microorganisms12040674

**Published:** 2024-03-28

**Authors:** Carlos Alfredo Carmona Gasca, Sergio Martínez González, Luz Olivia Castillo Sánchez, Ernesto Armando Rodríguez Reyes, María Fidelia Cárdenas Marrufo, Ignacio Vado Solís, Giselle Castañeda Miranda, Lilia Patricia López Huitrado, Alejandro de la Peña-Moctezuma

**Affiliations:** 1Functional Biology Laboratory, Academic Unit for Veterinary Medicine and Zootechnics, Autonomous University of Nayarit, Tepic 63155, Mexico; sergio.martinez@uan.edu.mx (S.M.G.);; 2Leptospira and Leptospirosis Research Group, Teaching, Research and Extension Center for Animal Husbandry in the Plateau, School of Veterinary Medicine and Zootechnics, National Autonomous University of Mexico, Tequisquiapan 76795, Mexico; 3Postgraduate Studies Division, School of Medicine, National Autonomous University of Mexico, Coyoacan 04510, Mexico; 4School of Medicine, Autonomous University of Yucatan, Merida 97000, Mexico; cmarrufo@correo.uady.mx (M.F.C.M.);; 5Academic Unit for Veterinary Medicine and Zootechnics, Autonomous University of Zacatecas, Zacatecas 98068, Mexico; lilia_patricia@uaz.edu.mx

**Keywords:** *Leptospira*, dog, carrier

## Abstract

Leptospirosis is a neglected zoonotic disease that commonly affects cattle, pigs, horses, and dogs in many countries. Infection in dogs is usually subclinical, but acute cases of leptospirosis may occur along with systemic failure, which may become fatal. After recovery from an acute infection, dogs may become asymptomatic carriers and shed pathogenic leptospires through urine for long periods of time. Here, a study of ten different cases of leptospirosis is presented, showing the relevance of dogs as asymptomatic carriers of pathogenic *Leptospira*. The diagnosis was confirmed via isolation and further serological and genetic identification. Four *Leptospira* isolates (LOCaS28, 31, 34, and 46) were obtained from the kidneys and urine samples of 58 dogs destined for destruction (6.89%) at a Canine Control Center in Mexico City. No spirochetes were observed in the urine samples of those *Leptospira*-positive dogs examined under dark-field microscopy, and no clinical signs of disease were observed either. Six additional isolates were obtained: two came from asymptomatic carrier dogs (CEL60 and UADY22); another isolate came from an asymptomatic dog that was a pack companion of a clinically ill dog with fatal leptospirosis (AGFA24); and finally, three isolates were taken from dogs that died of leptospirosis (LOCaS59, Citlalli, and Nayar1). Nine out of the ten isolates were identified as being from the serogroup Canicola via cross-absorption MAT using reference strains and specific antisera, and their identity was genetically confirmed as Canicola ST34 via multi-locus sequencing typing (MLST). In contrast, the isolate Nayar1 was identified as serovar Copenhageni ST2. Interestingly, the asymptomatic dogs from which *Leptospira* isolates were recovered consistently showed high antibody titers in the microscopic agglutination test (MAT), revealing values of at least 1:3200 against serogroup Canicola and lower titer values against other serogroups. Isolates showed different virulence levels in the hamster model. Taken as a whole, all these findings confirmed that dogs may act as asymptomatic carriers of pathogenic leptospires and possibly spread them out to the environment, thus representing an active public health risk. The results also showed that the Canicola ST34 clone is the most prevalent *Leptospira* serovar in dogs in Mexico, and finally that the old-fashioned MAT is a good alternative for the detection of presumptive *Leptospira* asymptomatic carrier dogs.

## 1. Introduction

Leptospirosis is an emerging infectious disease of global importance. It is caused by pathogenic spirochetes of the genus *Leptospira*, which currently includes 64 taxa, including 26 saprophytic and 38 pathogenic species [1,2]. Leptospirosis is considered one of the more widespread zoonoses worldwide [3]. Severe leptospirosis is related to the dysfunction of multiple organs (e.g., renal failure, severe hemorrhage, respiratory distress, and neurological damage), which can lead to death in 1–10% of cases [4]. After entering the system, the organism replicates in the blood vessels. Later, with the rising of specific circulating antibodies, the organism is hosted and maintained in the renal tubules and then chronically disseminated to the environment through urine.

Different wildlife species, including small rodents and domestic animal species, may act as both maintenance and incidental hosts of a variety of pathogenic *Leptospira* serovars. Domestic animals may act as maintenance hosts of adapted serovars, like Canicola in dogs, Hardjo in cattle, Pomona in horses, and Bratislava in pigs [5,6]. Rodents are also commonly infected with pathogenic *Leptospira* serovars but show no signs of infection. Rats usually act as reservoirs of the serovar Icterohaemorrhagiae, and mice as reservoirs of the serovar Ballum. The presence of such reservoirs is considered the main risk factor in human leptospirosis [7,8].

Companion and working dogs are continuously increasing in number among societies around the world, becoming the most common animal contact for humans, and so represent an important potential source of zoonotic diseases, including leptospirosis [9]. In a recent review, it was concluded that the more frequently detected *Leptospira* serovars in dogs are Canicola, Icterohaemorrhagiae, Grippotyphosa, and Pomona [10]. Dogs that acquire leptospirosis may either maintain an asymptomatic course of infection or present very diverse clinical presentations that range from mild clinical signs of disease through to severe and even fatal cases involving systemic infections with renal, liver, and lung failure [11,12]. Clinical signs of canine leptospirosis depend on the age and immunity of the host, environmental factors, and the virulence of the infecting serovar. Four clinical presentations of leptospirosis have been described and referred to as syndromes, including the icteric, hemorrhagic, uremic and reproductive syndromes [6]. In that respect, two consensus agreements on the clinical signs and laboratory findings that must be considered for the diagnosis of a case of canine leptospirosis have been established in recent years [11,12]. After acquiring a leptospiral infection, dogs may either maintain an asymptomatic state of infection or develop clinical leptospirosis. After recovery, dogs may then act as carriers of pathogenic leptospires, shedding the organism into environment, and so represent a health risk for owners, the public, other dogs, and other mammals [13]. In this study, we report on the recovery and characterization of pathogenic *Leptospira* from kidney and urine samples obtained from asymptomatic carrier dogs, as well as from dogs with clinical and fatal leptospirosis.

## 2. Materials and Methods

### 2.1. Leptospira Strains and Serological Diagnosis

Thirteen reference strains, representing four *Leptospira* species (*L. borgpetersenii*, *L. interrogans*, *L. kirschneri*, and *L. weilii*), were used in the microscopic agglutination test (MAT), which is the gold standard for the diagnosis of leptospirosis. Strains were kindly donated by the WHO/FAO/OIE-Collaborative Centre of Reference and Research on Leptospirosis, Australia and Western Pacific Region, Brisbane, Australia (Table 1). MAT was performed as described by Goris and Hartskeerl [14].

### 2.2. Samples Collection and Culture

In an open study conducted to identify *Leptospira* carriers, 58 dogs were sampled. These animals had no apparent signs of disease (asymptomatic) and were destined for destruction in a canine control centre in Mexico City. The sampled dogs were mixed-breed adults, 33 males and 25 females, with no apparent signs of illness. Each dog was anesthetized, and a blood sample obtained from the cephalic vein. The abdominal area was shaved and disinfected with a 1.5% iodine solution. Then, an abdominal longitudinal incision was performed to remove one kidney and to obtain a 5 to 10 mL urine sample via a direct puncture of the urinary bladder with a sterile syringe. The kidney was handled aseptically and immediately submerged into a 0.5% benzalkonium chloride solution to avoid any possible further contamination. After ten minutes disinfection, kidneys were dried with sterile paper towels and then cut longitudinally with a sterile surgical knife to expose the renal parenchyma. Approximately 500 mg of tissue was obtained from the renal medulla and cortex with a sterile scraper and then resuspended in 1 mL of EMJH liquid medium. A 0.3 mL sample of such a suspension was inoculated into 6 mL of EMJH medium. Three drops of the urine samples previously taken with a sterile syringe were directly inoculated into both EMJH and Fletcher media. Finally, three tenfold dilutions of both samples were conducted in EMJH liquid and Fletcher semisolid media and incubated at 30 °C.

Two additional *Leptospira* isolates, obtained from the kidneys of two asymptomatic dogs, were included in this study: CEL60 and UADY22 isolates. These were taken from dogs that had been included in two independent epidemiological studies. These studies were performed, respectively, in the city of Toluca in central Mexico (Carmona-Gasca, personal communication), and in the city of Merida, Yucatan, in southern Mexico (Cárdenas Marrufo and Vado Solís, personal communication). The isolate AGFA24 was obtained from another asymptomatic carrier dog; this individual was the companion of a dog with fatal icteric leptospirosis. A urine sample was taken after the detection of a titer as high as 1:6400 against serogroup Canicola in the MAT. The urine sample from this suspicious carrier dog was obtained via cystocentesis, practiced at the veterinary hospital, and cultured immediately in EMJH and Fletcher media.

In the case of dogs that died of clinical leptospirosis, blood, kidney and urine samples were aseptically collected and handled as described. In any case, cultures were incubated at 30 °C for up to 12 months before being discarded.

### 2.3. DNA Extraction

Isolates were purified via the “maximum dilution” method. This method consists in a series of tenfold dilutions of the cultures up to the maximum dilution at which leptospiral growth is obtained, usually up to a dilution of 10^15^. Then, the purified isolate was cultured in 100 mL of EMJH medium at 30 °C for 7 to 10 days. Well-developed cultures were concentrated via centrifugation at 10,000× *g* for 30 min and DNA was extracted from the pellet via the DNA was extracted from the pellet via the guanidine thiocyanate and isobutyl alcohol method, as described previously [15].

### 2.4. PCR, Purification, and Sequencing of Amplicons

The PCR amplification of gene fragments for MLST characterization [16] was performed in a 50 µL volume reaction, using 200 mM of dNTPs (Roche^®^, Indianapolis, IN, USA), 5 µL 10× buffer, 10 pmol of each primer, and 30 to 60 ng of genomic DNA as a template. All primers were synthesized at the Oligonucleotide Synthesis Facility, Biotechnology Institute, National Autonomous University of Mexico (UNAM), Cuernavaca, Morelos, Mexico. Reactions were performed in a Perkin Elmer 2400 PCR System thermocycler (Perkin Elmer, Waltham, MA, USA). The PCR parameters for gene amplification were as follows: an initial denaturation step at 95 °C for 5 min, followed by 40 cycles of denaturation at 94 °C for 30 s, alignment at 54 °C for 45 s, extension at 72 °C for 45 s, and one final extension cycle at 72 °C for 7 min. PCR products were separated via electrophoresis, visualized in ethidium bromide-stained agarose gels, and purified using the Montage^®^ DNA purification columns. Amplicons were sequenced at the Biotechnology Institute, UNAM, using an Applied Biosystems 3730 device (Applied Biosystems, Waltham, MA, USA) with the Taq Dye Terminator Cycle method, as well as in the Macrogen Inc. facility, Seoul, Korea using an Automatic Sequencer 3730XL.

### 2.5. Sequence Editing and Analysis

Chromatograms quality and sequences editing were performed with the Sequencher 4.6 program (Gene Codes Corporation, Ann Arbor, MI, USA). The characterization of isolates was performed using internal fragments of the genes proposed in the schemes of Ahmed et al. [16], including *adk* (adenylate kinase), *icdA* (isocitrate dehydrogenase), *lipL32* (outer membrane lipoprotein LipL32), *lipL41* (outer membrane lipoprotein LipL41), *rrs2* (16S rRNA), and *secY* (pre-protein translocase SecY). The sequences obtained from each gene fragment were compared using the MLST database available at https://pubmlst.org/organisms/leptospira-spp (accessed on 31 January 2024). Allelic numbers were determined using the public database for molecular typing and the assessment of microbial genome diversity, PubMLST (scheme#3). The analysis of phylogeny was conducted using the *rrs* sequences taken from dog *Leptospira* isolates, the *rrs* sequences of eleven reference strains, and the Genebank sequences. The phylogeny analysis was conducted with the Molecular Distance matrix program using the neighbor-joining method [17]. This included examining 1000 replicates in the bootstrap test with Evolutionary Genetics Analysis version 11 (MEGA 11) [18]. Finally, the Mann–Whitney test for independent samples was used to determine the relationship between an antibody titer ≥ 1:3200 and the isolation of *Leptospira*, and a *p* ≤ 0.05 was considered significant.

### 2.6. Virulence Quantification

Virulence of the purified isolates was tested in the hamster model (*Mesocricetus auratus*). For isolates LOCaS59, AGFA24, and Nayar1, a 0.2 mL volume from a solution containing 2 × 10^4^ leptospires per mL was intraperitoneally inoculated in at least two hamsters per isolate. A higher dose up to 4 × 10^5^ was used for isolates LOCaS28, 31, and 34. The LD50 (letal dose 50%) was calculated using the Reed–Muench method for the isolates LOCaS46 and Citlali, using three groups of ten hamsters for each isolate.

## 3. Results

### 3.1. MAT and Leptospira Isolates

Overall, 35 out of the 58 asymptomatic dogs (60.3%) showed antibody titers of at least 1:100 against *Leptospira*: 18 were males (51.4%) and 17 were females (48.5%). The most frequently detected serogroups in the MAT were Canicola (23, 39.6%), Bratislava (20, 34.4%), Pyrogenes (19, 32.7%), Grippotyphosa (11, 18.9%), and Icterohaemorrhagiae (7, 12%). In total, 10 out of the 58 dogs (17%) (7 males and 3 females) showed antibody titers ≥ 1:3200. Four *Leptospira* isolates (LOCaS28, 32, 34, and 46) were obtained from the 58 asymptomatic sampled dogs (6.89%). All of them were male dogs and all of them showed a titer ≥ 1:3200 against serogroup Canicola. Isolates were obtained in a Fletcher medium after incubation at 30 °C for periods ranging from 16 up to 56 weeks. A direct association as high as 1:3200 or greater was observed between the isolation of *Leptospira* and the presence of antibody titers in asymptomatic dog carriers (*p* > 001). As expected, a direct relationship was found between the isolated serovar and the highest antibody titer in the MAT against the corresponding serogroup (Table 2). However, that finding was not consistent in the dogs with clinical leptospirosis strains LOCaS59 and Nayar1. Three more *Leptospira* isolates were obtained from asymptomatic carrier dogs in independent studies (CEL60, UADY22, and AGFA24). Finally, another three isolates came from dogs that died from clinical leptospirosis (LOCaS59, Citlali, and Nayar1) (Table 2).

### 3.2. Genetic Diversity of the Infecting Leptospira isolates

Serologic identification confirmed that all isolates, except for Nayar1, belonged to serogroup Canicola, including those obtained from asymptomatic dogs or from dogs with clinical leptospirosis. Nayar1 was the only isolate identified differently as belonging to serogroup Icterohaemorrhagiae.

DNA was obtained from all ten isolates, and a 16S rRNA gene amplicon was generated via PCR and sequenced. Overall, 16S rRNA phylogenetic studies showed that all isolates belong to the species *L. interrogans* (Figure 1).

Further MLST characterization confirmed the identity of the isolates. The dog isolates in this study, identified as belonging to the serovar Canicola, were obtained from different regions of Mexico, including Mexico City, Yucatan, Toluca, and Queretaro. These isolates were obtained from asymptomatic carrier dogs (seven) or from sick dogs (two), and all of them showed the same allelic profile, corresponding to the sequence type (ST) ST34 (*secY*6, *adk*2, *rrs*2, *lipL32*1, *lipL41*1, or *icdA*1). ST34 is also the corresponding ST for the reference strain *L. interrogans* of the serovar Canicola Hond Utrecht IV. In contrast, a dog isolate obtained from a fatal hyperacute case of leptospirosis in a puppy in Tepic City in the State of Nayarit (Nayar1) showed the allelic profile ST2, corresponding to serovar Copenhageni in the *L. interrogans* serogroup Icterohaemorrhagiae. Experimental infection in the hamster model, which was performed with some of the *Leptospira* isolates obtained from the dogs, showed different levels of virulence (Table 3).

## 4. Discussion

A very important characteristic to be considered for the diagnosis of leptospirosis is the diversity of clinical signs that might be observed in both human and veterinary practice. The severity of the disease may range from a mild influenza-like illness through to a severe syndrome characterized by jaundice, renal failure, myalgia–arthralgia, and myocarditis known as Weils’s disease. A clinical picture of meningitis and severe, often fatal pulmonary hemorrhaging are also characteristic clinical presentations of leptospirosis [19]. In dogs, at least four leptospirosis syndromes have been described. These include typical icteric syndrome, with fever, jaundice, and renal failure as the main clinical signs; uremic syndrome, also known as Stuttgart disease, which is characterized by vomiting and the presence of oral and gastrointestinal ulcers as the result of acute renal failure; hemorrhagic syndrome, with an acute outcome of bloody diarrhea, vomiting, dehydration, shock, and death; and reproductive syndrome, characterized by infertility, abortion, and the premature birth or perinatal death of weak puppies [6]. Such a diversity of clinical presentations has been related to the level of adaptation of some *Leptospira* serovars to different animal species. In the three clinical cases reported in this paper, the two adult dogs presenting an icteric syndrome (LOCaS59 and Citlali) were both infected with the serovar Canicola ST34; meanwhile, serovar Copenhageni ST2 (Nayar1) was isolated from a three-month-old puppy that died of a highly acute case of leptospirosis. It has been observed that host-adapted serovars usually cause chronic and subclinical infections, while non-adapted serovars are usually involved in acute, severe, and often fatal cases of leptospirosis [5]. In dogs, host-adapted *Leptospira* serovars such as Canicola usually cause a subclinical form of leptospirosis, and a carrier state might be established with no apparent clinical signs [20].

The isolation of *Leptospira* from clinical samples is usually not an easy task. However, it is advisable to attempt to obtain such isolates for the improvement of epidemiological data collection, as well as for the development of more effective immunogens, at least until better vaccines become available [6,21,22]. Furthermore, there is a preference for using local isolates as more specific antigens in the MAT for the serological diagnosis of leptospirosis [21].

The serovar Canicola is considered endemic and is highly adapted to dogs. Dogs usually show a high frequency of infection with this serovar but, on the other hand, dogs also show a higher resistance against the virulence of serovar Canicola. In contrast, infection with the serovar Icterohaemorrhagiae, the rat-adapted serovar, is usually associated with a more aggressive outcome of leptospirosis. This partially explains why the serovar Canicola is usually associated with persistent renal colonization and chronic and inapparent subclinical infection in dogs [22,23]. On the other hand, it has been observed that exposure to non-adapted serovars is usually associated with acute and often fatal disease and a high antibody response [22]. Nevertheless, in the present study, we observed a very high antibody response (1:3200 or higher) in 10 out of the 58 asymptomatic dogs (17.2%), and four *Leptospira* isolates were obtained from them (6.89%). The serovar Canicola was isolated from the kidney tissue or urine samples of those four asymptomatic carrier dogs (LOCaS 28, 31, 34, and 46). This was also observed in the dogs from which isolates UADY22 and AGFA24 were obtained, and whose antibody titers were, respectively, as high as 1:3200 and 1:6400 against serogroup Canicola in the MAT. The isolate UADY22 was obtained from an asymptomatic stray dog from Merida, in the southern Mexican state of Yucatan. The dog showed an MAT titer of at least 1:3200 against serogroup Canicola. Such an isolate was obtained in parallel with the performance of a serological and pathological study in dogs reporting a high frequency of renal lesions as well as 34% seropositivity, mainly against serogroup Canicola [24]. On the other hand, isolate AGFA24, also characterized as the serovar Canicola ST34, was recovered from a urine sample collected via cystocentesis and cultured in Fletcher semisolid medium after 38 days of incubation at 30 C. The dog from which the isolate AGFA24 was recovered was an asymptomatic carrier dog, which was a pack companion of a sick animal that died of an acute icteric syndrome of leptospirosis. PCR of a urine sample of this dog detected leptospiral DNA. Despite the high serum antibody titer, this dog did not show any clinical signs of disease and its blood chemistry parameters used to evaluate hepatic and renal function were normal. Furthermore, a 15-day oral administration of 8 mg/kg of doxicicline b.i.d. resulted in negative PCR results in urine samples and a reduction in MAT titers against serogroup Canicola, which fell from 1:6400 to 1:800 after treatment. These findings contradict the belief that high antibody titers are related always to an active case of clinical leptospirosis in dogs [11,12,23]. In contrast, the asymptomatic carrier dog from which the isolate CEL60 was recovered showed an MAT antibody titer as low as 1:200 against serogroup Canicola.

On the other hand, isolates LOCaS59 and Citlali were recovered from renal tissues obtained from two adult dogs with acute icteric syndrome of leptospirosis. These two isolates were characterized. We also characterized the serovar Canicola ST34, which was the same ST as that possessed by the isolates obtained from asymptomatic carrier dogs. These findings show that the dog-adapted serovar Canicola could either be present in the renal tissue of asymptomatic carrier dogs, or become a pathogen associated with multiorgan dysfunction and death in susceptible dogs. Some predisposing factors, such as immunocompromise and the virulence of the infecting organism, could lead to such often-fatal and severe cases of leptospirosis in dogs.

Non-adapted serovars in dogs, such as Icterohaemorrhagiae and Copenhageni, are usually hosted by rodents acting as reservoirs [6]. It has been reported that infection with these serovars includes symptoms of fever, congested mucous membranes, depression, anorexia, hemorrhages, oliguria, anuria, renal pain, myalgia, bloody diarrhea, jaundice, and dyspnea [23]. The isolate Nayar1, reported in this study, was identified via MLST as serovar Copenhageni ST2. This isolate was recovered from a blood sample of a three-month-old bull terrier male with an acute onset of the hemorrhagic syndrome of leptospirosis. At physical examination, the puppy was hypothermic and presented fetid diarrhea, emesis, abdominal tenderness, and jaundice. There was no history of previous vaccination against leptospirosis. The isolate was recovered in Fletcher medium after 40 days of incubation at 30 °C. The recovery of this *Leptospira* isolate from blood showed that, in this hyperacute case of leptospirosis, the patient was in the leptospiremic phase of the infection at the time of the sample collection. The patient finally died 8 h after the arrival at the clinic and the performance of sample collection. It has been reported that, in clinical leptospirosis, specific antibodies are detected between days 6 to 10 post-infection [25]. So, in the early stages of infection, agglutinating antibodies are not detected by the MAT. This is the reason why it is advisable to obtain a paired serum sample, collected 15 days after the first sample. This practice allows the detection of a rise in serum antibody titers, revealing an active case of leptospirosis [19]. In the case of the dog from which the serogroup Copenhageni was isolated (Nayar1), the patient showed an antibody titer of 1:3200 against serogroup Canicola, but only 1:200 against serovar Icterohaemorrhagiae in the MAT. This low titer against the isolated serovar (Copenhageni belongs to the Icterohaemorrhagiae serogroup) might well be related to the hyperacute onset of the disease, which gives no time for a rise in the antibody titer. In addition, it is possible that, due to the severe acute infection, urinary density was within normal parameters, contrary to what is usually reported in subacute cases of canine leptospirosis [26]. However, the high antibody titer observed against serogroup Canicola in the absence of the isolation of such a serovar remains obscure. Potential explanations for this include the possibility of a cross-reaction between both serovars, or the presence of maternally derived antibodies [27].

The virulence of the isolates was tested in the hamster model. The signs and lesions observed in the infected specimens, particularly severe pulmonary hemorrhages and death, demonstrated the virulence of such *Leptospira* isolates. The virulence of those isolates varied from very high (LOCaS 46: LD50 ≤ 4), to high (Citlali: LD50 = 25), to very low (LOCaS 31 and 34 LD50 > 40,000) (Table 3).

One of the most accurate approaches for the typification of *Leptospira* isolates is MLST (multilocus sequencing typing). MLST utilizes a series of highly variable but conserved genes whose sequences variations allow researchers to obtain a quite accurate and specific sequence type (ST). Such an ST is associated with a specific serovar and allows for the identification of isolates globally [16,28]. MLST allowed for the identification of every Canicola isolate in this study as Canicola ST34, and of the Copenhageni isolate as ST2 [16].

It is well known that rodents represent the main predisposing factor for human leptospirosis [25]. However, the presence of *Leptospira* carrier dogs represents a potential risk to public health. As such, the detection of dogs acting as carriers must be considered an important issue in veterinary practice. The recovery of *Leptospira* virulent isolates from at least six asymptomatic dogs in this study highlights the importance of applying measures from preventive medicine to avoid the risk of transmission to owners or other dogs. Furthermore, the fact that such a confirmed carrier state of leptospirosis in those six dogs was closely related to a high serum antibody titer, detected using the MAT (1:3200 or higher), suggests the value of applying this simple test to detect highly suspicious *Leptospira* carrier dogs. This finding has scarcely been discussed in the literature [29]. High antibody titers with uses against *Leptospira* are usually caused by an active infection and disease, or sometimes by a recent vaccination [30]. However, our findings show that dogs acting as carriers and potential disseminators of virulent leptospires might reach high-MAT titers against infecting *Leptospira* serovars. The *Leptospira* carrier dogs detected in this study consistently showed high antibody titers against the *Leptospira* serogroup, but no signs of illness. Also, variable degrees of virulence between such isolates were observed in the hamster model of infection. All these results suggest that, in a routine visit to the veterinary clinic, it might well be advisable to collect a blood sample in order to test for immunologic profiles of zoonotic diseases such as leptospirosis, particularly when risk factors such as the presence of rodents or open water sources are part of the living conditions of companion dogs [6,25,31,32].

Canine leptospirosis is an important zoonotic disease in many countries. Disease transmission usually occurs from infected hosts via urine-contaminated environmental sources, such as water. Direct contact between infected and susceptible individuals, environmental factors such as climate-related changes in temperature and/or rainfall, and increasing numbers of rodents, acting as reservoirs, may increase dog exposure risks. A dog’s lifestyle may influence exposure risk to leptospirosis, but vaccination based on the proper identification of circulating *Leptospira* serogroups dramatically reduces post-exposure infections [32]. Regrettably, resistance to vaccination by veterinarians and dog owners leaves many dogs at risk of this zoonotic disease.

## Figures and Tables

**Figure 1 microorganisms-12-00674-f001:**
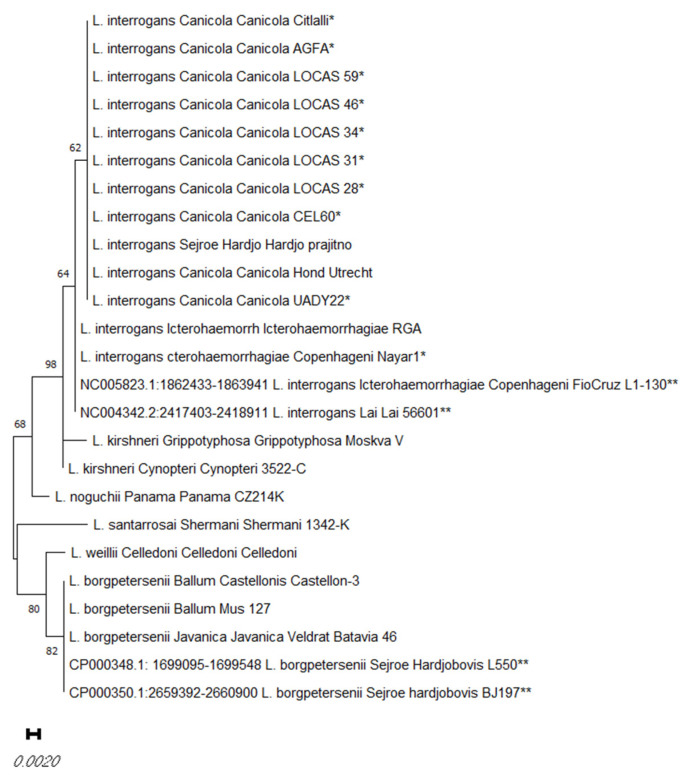
Distance matrix of the selected *rrs* sequences obtained from the dog *Leptospira* isolates in this study (*), and the *rrs* sequences of reference strains and Genebank sequences (**). The distance matrix was obtained via the neighbor-joining method. The percentage of replicate trees in which the associated taxa clustered together in the bootstrap test (1000 replicates) is shown next to the branches.

**Table 1 microorganisms-12-00674-t001:** *Leptospira* reference strains used in the microscopic agglutination test (MAT).

Serovar	Serogroup	Strain	Species
Australis	Australis	Ballico	*L. interrogans*
Autumnalis	Autumnalis	Akiyami A	*L. interrogans*
Bataviae	Bataviae	Van Tienen	*L. interrogans*
Bratislava	Australis	Jez Bratislava	*L. interrogans*
Canicola	Canicola	Hond Utrecht IV	*L. interrogans*
Celledoni	Celledoni	Celledoni	*L. weilii*
Grippotyphosa	Grippotyphosa	Moskva V	*L. kirschneri*
Hardjo	Sejroe	Hardjoprajitno	*L. interrogans*
Icterohaemorrhagiae	Icterohaemorrhagiae	RGA	*L. interrogans*
Pomona	Pomona	Pomona	*L. interrogans*
Pyrogenes	Pyrogenes	Salinem	*L. interrogans*
Tarassovi	Tarassovi	Perepelitsin	*L. borgpetersenii*
Wolffi	Sejroe	3705	*L. interrogans*

**Table 2 microorganisms-12-00674-t002:** Epidemiological, clinical, and serological findings of dogs from which leptospires were isolated.

Isolate	Isolation		Dog Sex/Age/Breed	Clinical Condition	MAT Titer	Source of the Isolate	Isolate Identification
	Year	Place					
CEL60	2002	Toluca, Mexico State	M/adult/mixed	Asymptomatic	Canicola 1:200 Icterohaemorrhagiae 1:200 Pyrogenes 1:50	Kidney	Canicola
LOCaS28	2004	Mexico City	M/adult/mixed	Asymptomatic	Bratislava 1:800 Canicola 1:3200 Pyrogenes 1:6400	Urine	Canicola
LOCaS31	2004	Mexico City	M/adult/mixed	Asymptomatic	Canicola 1:3200 Icterohaemorrhagiae 1:200 Pyrogenes 1:1600	Kidney	Canicola
LOCaS34	2004	Mexico City	M/adult/mixed	Asymptomatic	Canicola 1:3200 Pyrogenes 1:200 Icterohaemorrhagiae 1:50	Kidney	Canicola
LOCaS46	2004	Mexico City	M/adult/mixed	Asymptomatic	Bratislava 1:1600 Canicola 1:6400 Pyrogenes 1:3200	Kidney	Canicola
UADY22	2002	Merida, Yucatan	M/adult/mixed	Asymptomatic	Canicola 1:3200 Icterohaemorrhagiae 1:400 Grippotyphosa 1:400	Kidney	Canicola
AGFA24	2017	Ezequiel Montes, Queretaro	M/7 years/mixed	Asymptomatic	Bataviae 1:800 Canicola 1:6400 Celledoni 1:800	Kidney	Canicola
LOCaS59	2004	Mexico City	M/adult/mixed	Icteric syndrome	Canicola 1:100 Icterohaemorrhagiae 1:800 Pyrogenes 1:3200	Kidney	Canicola
Citlalli	2011	Calamanda, Queretaro	M/adult/dachshund	Icteric syndrome	Canicola 1:6400 Grippotyphosa 1:3200 Icterohaemorrhagiae 1:800	Kidney	Canicola
Nayar1	2019	Tepic, Nayarit	M/3 months/bull terrier	Hyperacute icteric syndrome	Canicola 1:3200 Grippotyphosa 1:100 Icterohaemorrhagiae 1:200	Blood	Copenhageni

MAT: Microscopic agglutination test; Relationship between ≥1:3200 titers in the MAT and isolation of *Leptospira* (*p* > 0.0002).

**Table 3 microorganisms-12-00674-t003:** Serologic and genetic identification of the dog isolates and virulence in the hamster model.

Isolate	MAT Identity	16S rRNA Identity	MLST Profile							Virulence to Hamster	
			*adk*	*icdA*	*lipL32*	*lipL41*	*rrs2*	*secY*	ST	Dead (%)	Days to Dead
CEL60	Canicola	*L. interrogans*	2	1	1	1	2	6	34	ND	ND
LOCaS28	Canicola	*L. interrogans*	2	1	1	1	2	6	34	2/9 (22.2)	7.25
LOCaS31	Canicola	*L. interrogans*	2	1	1	1	2	6	34	2/4(50) ^A^	6.5
LOCaS34	Canicola	*L. interrogans*	2	1	1	1	2	6	34	2/4 (50) ^A^	30
LOCaS46	Canicola	*L. interrogans*	2	1	1	1	2	6	34	9/10 (90) ^B^	5.44
UADY22	Canicola	*L. interrogans*	2	1	1	1	2	6	34	ND	ND
AGFA24	Canicola	*L. interrogans*	2	1	1	1	2	6	34	2/2 (100)	5
LOCaS59	Canicola	*L. interrogans*	2	1	1	1	2	6	34	2/4 (50)	8
Citlalli	Canicola	*L. interrogans*	2	1	1	1	2	6	34	6/10 (60) ^C^	12
Nayar1	Copenhageni	*L. interrogans*	1	1	2	2	1	1	2	2/2 (100)	5

MAT: Microscopic agglutination test; MLST: Multiple locus sequencing typing; ND: Not done; ^A^: LOCaS31 and 34 lethal dose = >40,000; ^B^: LOCaS46 LD50 = ≤4; ^C^: Citlali LD50 = 25.

## Data Availability

All data is available on request to Alejandro de la Peña-Moctezuma delapema@unam.mx and Carlos alfredo Carmona Gasca carmonagasca@uan.edu.mx.

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
