# Peer review of "The Presence of a Virulent Clone of Leptospira interrogans Serovar Canicola in Confirmed Cases of Asymptomatic Dog Carriers in Mexico"

_microorganisms, 2024, doi:10.3390/microorganisms12040674_

Round 1

Reviewer 1 Report

Comments and Suggestions for Authors

microorganisms-2852817

A virulent clone of Leptospira interrogans serovar Canicola is present in confirmed cases of asymptomatic dog carriers in Mexico

Carmona Gasca et al

This study provides valuable information on asymptomatic carriage of pathogenic Leptospira interrogans among dogs. Shedding from these animals may infect susceptible hosts including dogs and other species including humans. The seven asymptomatic samples were obtained from a variety of sources and this information was supplemented by samples from three canine cases which had died from leptospirosis. The introduction provided sufficient background information and was presented logically. The methods and results were missing some details (see below) but this should be simple to fix. Likewise, the discussion could benefit from a little more emphasis on the Canicola serovar and potential zoonotic transmission among hosts especially dogs and rodents. One aspect that I found difficult while reading this paper was the punctuation use, particularly the overuse of commas. In my view, this work would benefit from some copy editing before publication to improve the flow and clarify the text. Overall though I found the study to be of interest, methodologically sound and clearly presented apart from a few omissions of details (see below). Included below are some specific comments and suggestions which I hope the authors will find useful.

Line 60 – Change representing to represent.

Line 82 – There seems to be a difference in the text on lines 81-83 and the information in table 1. The Lepto species L. noguchii and L. santarosai are not in the table (only 4 are presented). Also, there are 13 reference strains in the table (vs 12 in text).

Line 106 - It is not clear how the samples from the three asymptomatic cases which were not from the canine control centre (CEL60, UADY22 and AGFA24) were obtained and cultured. In table 3 it indicates that all isolates were from kidney samples – were other sources investigated? How and why were these animals selected for study.

Line 113 – I’m not familiar with maximum dilution. Consider clarifying.

Line 137-8 – consider moving the sentence about the primers up to line 120 with other PCR details.

Line 145-147 – More detail is needed for the corelation analysis. It is not clear what was tested.

There is no information in the methods on the hamster model virulence tests.

Line 178 – figure 1 also differentiates the nine Canicola samples from the (Nayar1) sample using 16S. Figure was a little confusing. The figure is an NJ tree not a distance matrix. There is an * next to a sample (Portland vere) which does not seem to be one of the study samples. The ** samples should have accessions. It is unclear in the caption where the remaining sequences used in the analysis came from.

Discussion first paragraph. Im my view it is worth noting when discussing the different disease outcomes that the icteric symptoms were seen in the canicola isolates while the hemorrhagic symptoms in the different, Copenhagi isolate. Nayar1 is discussed later but it is relevant to show different symptoms from different isolates here as well.

Lines 295-305 - Regarding the lower MAT titer reactions from multiple isolates –in addition to the factors which are discussed, is cross reactivity a factor here?

Last paragraph of discussion might include some information on zoonotic risk specific to the canicola serovar. For instance, what are potential transmission links between dogs and rodents? Previous studies in other hosts e.g. cattle, human to highlight transmission risks.

Comments on the Quality of English Language

See comments above regarding punctuation. There were a few other minor grammatic errors throughout e.g. line 47 (change to organ dysfunctions) and line 60 (change representing to represent). These are relatively minor and will hopefully be easy to fix. 

Author Response

Thank you, we do really appreciate the reviewers comments.

Answers to comments from Reviewer 1.

Line 60 – Change representing to represent.

Thank you, we have fixed this word, now in line 65.

Line 82 – There seems to be a difference in the text on lines 81-83 and the information in table 1. The Lepto species L. noguchii and L. santarosai are not in the table (only 4 are presented). Also, there are 13 reference strains in the table (vs 12 in text).

We have fixed that information and now corresponds with that in Table 1.

Line 106 - It is not clear how the samples from the three asymptomatic cases which were not from the canine control centre (CEL60, UADY22 and AGFA24) were obtained and cultured.

A paragraph explaining this has been included in lines 114 – 120.

In table 3 it indicates that all isolates were from kidney samples – were other sources investigated? How and why were these animals selected for study.

In Table 2, it is shown that isolate LOCaS28 was obtained from a urine sample. Urine samples were also investigated, as it is said in lines 103 – 104.

Line 113 – I’m not familiar with maximum dilution. Consider clarifying.

A brief description of the method has been included in lines 128 – 130.

Line 137-8 – consider moving the sentence about the primers up to line 120 with other PCR details.

Primers information has been included with the PCR details on lines 136 – 148.

Line 145-147 – More detail is needed for the correlation analysis. It is not clear what was tested.

 An explanatory paragraph has been included in lines164 – 166. “Finally, to determine the relationship between an antibody titre ≥ 1:3,200 and the isolation of Leptospira, the Mann-Whitney test for independent samples was used, a p< 0.05 was considered significant.”

There is no information in the methods on the hamster model virulence tests.

A short paragraph describing virulence tests in hamsters has been included. Lines 168 – 173.

Line 178 – figure 1 also differentiates the nine Canicola samples from the (Nayar1) sample using 16S. Figure was a little confusing. The figure is an NJ tree not a distance matrix. There is an * next to a sample (Portland vere) which does not seem to be one of the study samples. The ** samples should have accessions. It is unclear in the caption where the remaining sequences used in the analysis came from.

The Figure has been edited, now includes the required information.

Discussion first paragraph. Im my view it is worth noting when discussing the different disease outcomes that the icteric symptoms were seen in the canicola isolates while the hemorrhagic symptoms in the different, Copenhagi isolate. Nayar1 is discussed later but it is relevant to show different symptoms from different isolates here as well.

A paragraph about the icteric clinical presentation and the hemorrhagic presentation and the serovars involved on each of them is presented in lines 258 – 261.

Lines 295-305 - Regarding the lower MAT titer reactions from multiple isolates –in addition to the factors which are discussed, is cross reactivity a factor here?

A short explanation has been included as well as a supporting reference [27] in lines 334 – 336.

Last paragraph of discussion might include some information on zoonotic risk specific to the canicola serovar. For instance, what are potential transmission links between dogs and rodents? Previous studies in other hosts e.g. cattle, human to highlight transmission risks.

Thank you, a paragraph has been included in lines 368 – 377.

Reviewer 2 Report

Comments and Suggestions for Authors

Overview: In this prospective study, the authors collected kidney tissue and/or urine from stray dogs that were asymptomatic and intended for destruction, owned dogs that were asymptomatic and "suspicious for being carriers" or dogs that died from leptospirosis. The high points of this study is the isolation and identification of serovar Canicola from all but one dog (Copenhageni in that dog) that was pathogenic in at least some of the dogs.

The low point of this study is that the authors do not present any evidence that the "asymptomatic" dogs were, in fact, asymptomatic. All of these dogs could have been symptomatic for leptospirosis. It seems more likely that the authors have identified a particularly virulent strain of serovar Canicola that is minimally host-adapted to dogs. The authors do not present any data to suggest that any dog in this study was a long-term carrier/shedder of this serovar. Correction of the manuscript to address this shortcoming is required.

Specific comments:

Line 69: The diagnosis is based on specific test results, not a combination of clinical signs and lab findings.

Line 82: “microscopic” not “microscopy”

Line 95: Poor sentence structure—what exactly does one do to prepare an abdomen for surgery other than shave and disinfect?

Line 98-: If the kidney is aseptically handled, why does it need to be subsequently disinfected in benzalkonium chloride?

Line 101-: What kidney tissue was obtained? Just the cortex as seems implied by the previous sentence? How much kidney tissue is obtained? What’s the final volume when resuspended in 1-ml of EMJH?

Line 102-: How are these ten-fold dilutions accomplished?

Line 104-: What’s the purpose of 1, 2 and 3 drops of urine in both EMJH and Fletcher media? This does not sound like a standard protocol.

Line 111: “being” not “been”

Line 114: “EMH” or “EMJH”

Line 166-167: This makes no sense. Even 167-170 are poorly worded.

Line 173: “except for” not “but”

Line 231-232: “vomiting” not “vomit”

Line 236: no comma

Line 242: remove “quite”

Line 249: “rat-adapted” not “rats adapted”

Line 256: “kidney” not “kidneys”

Line 273-274: Poor grammar. How is an aggressive pathogen different from a pathogen? I would remove the word “aggressive”.

Line 275-277: This is speculative. It is not stated what dose of steroids the dog was receiving. For example, a dog with hypoadrenocorticism receives prednisone every day for years, but that dose is not immunosuppressive. An anti-inflammatory dose of prednisone is also not immunosuppressive. This sentence should be removed and the prior statement about the dog clarified (dose (mg/kg/day) if possible or amended.

Line 283-303: The authors speculate about the reason this puppy had a reciprocal titer of 3200 to serovar Canicola and a reciprocal titer of 200 to serovar Icterohaemorrhagiae when the dog was infected with serovar Copenhageni. The authors proposed theory that the dog was co-infected with both serovars is illogical. Natural infection confers good cross-serovar protective immunity and the titer to serovar Canicola would suggest a staggered infection, not co-infection. Even then, establishment of a second infection would seem unlikely. It’s possible that the Canicola titer represents maternal antibody from vaccination or that the dog was previously vaccinated for serovar Canicola (although one would imagine the dog was vaccinated for Icterohaemorrhagiae, too).

Line 301: What does “…no decrease in the urine density” mean? There is no context to understand this statement.

Line 321- (and abstract): This study did not compare the serum titer to Canicola for dogs not shedding leptospires, so stating that using the MAT has value in identifying carriers is misleading. Likewise, I think it is inaccurate to characterize the dogs intended for destruction as “asymptomatic”. How long were these dogs held by the Canine Control Centre? What veterinary evaluation was performed? There was not any bloodwork (CBC, serum biochemistry profile) performed on these dogs, so stating that they were asymptomatic is potentially a gross mischaracterization. The same can be said of the three “suspicious carrier dogs” from which only urine was collected. If not laboratory assessment of these dogs was performed, how can it be stated definitively that they were healthy dogs (or at least asymptomatic)?

Line 326-: The authors cannot state that their dogs were “acting as carriers and disseminators of virulent leptospires”.  That potential may exist, but they don’t know that definitively.

Line 331-: This is a poorly worded sentence, but also complete nonsense. For starters, this study has nothing to do with ehrlichiosis and it’s unclear why the authors now mention it. Do the authors suggest routine MAT testing of every dog every year? Since the authors didn’t do MAT testing on the 54 dogs that were destroyed that didn’t have leptospires identified in the kidney, they can’t make any statement as to the value of annual MAT testing. No other study to date would support that suggestion.

Comments on the Quality of English Language

I tried to point out some of the problems with grammar in my comments above, but there are numerous that need to be cleaned up.

Author Response

Answers to comments from Reviewer 2.

Line 69: The diagnosis is based on specific test results, not a combination of clinical signs and lab findings.

The paragraph has been edited in this way: “In that respect, two consensus agreements on the clinical signs and laboratory findings that must be considered for the diagnosis of a case of canine leptospirosis, have been established in recent years [11-12].” Lines 75 – 77.

Line 82: “microscopic” not “microscopy”

Thank you, this word has been changed. Lines 87 and 93.

Line 95: Poor sentence structure—what exactly does one do to prepare an abdomen for surgery other than shave and disinfect?

It has been changed to this: “The abdominal area was shaved and disinfected with a 1.5% iodine solution.” Lines 100 – 101.

Line 98-: If the kidney is aseptically handled, why does it need to be subsequently disinfected in benzalkonium chloride?

Explained as: “The kidney was aseptically handled, and immediately submerged into a 0.5% benzalkonium chloride solution to avoid any possible further contamination.” Lines 103 - 105

Line 101-: What kidney tissue was obtained? Just the cortex as seems implied by the previous sentence? How much kidney tissue is obtained? What’s the final volume when resuspended in 1-ml of EMJH?

This was explained in this way: “Approximately 500 mg of tissue from the renal medulla and cortex were obtained with a sterile scraper and resuspended in 1 ml of EMJH liquid medium.” Lines 107 – 109.

Line 102-: How are these ten-fold dilutions accomplished?

Here the explanation: “A 0.3 ml sample of such a suspension was inoculated into 6 ml of EMJH medium. Three drops of the urine samples previously taken with a sterile syringe were directly inoculated into both, EMJH and Fletcher media. Finally, three tenfold dilutions of both samples were done in EMJH liquid and Fletcher semisolid media and incubated at 30 C.” Lines 109 – 113.

Line 104-: What’s the purpose of 1, 2 and 3 drops of urine in both EMJH and Fletcher media? This does not sound like a standard protocol.

Already fixed and explained in line 110-111: “Three drops of the urine samples previously taken with a sterile syringe were directly inoculated into both, EMJH and Fletcher media.”

Line 111: “being” not “been”

Fixed in line 125.

Line 114: “EMH” or “EMJH”

Fixed, thank you.

Line 166-167: This makes no sense. Even 167-170 are poorly worded.

This paragraph was changed in this way: “As expected, a direct relationship was found between the isolated serovar and the highest antibody titer in the MAT against the corresponding serovar (Table 2). Three more Leptospira isolates were obtained from carrier dogs in independent studies (CEL60, UADY22, and AGFA24). Finally, another three isolates came from dogs that died from clinical leptospirosis (LOCaS59, Citlali, and Nayar1)” Lines 181 – 185.

Line 173: “except for” not “but”

Changed, thank you. Line 194

Line 231-232: “vomiting” not “vomit”

Changed, thank you. Lines 253 and 255.

Line 236: no comma

Fixed

Line 242: remove “quite”

Fixed

Line 249: “rat-adapted” not “rats adapted”

Fixed

Line 256: “kidney” not “kidneys”

Fixed

Line 273-274: Poor grammar. How is an aggressive pathogen different from a pathogen? I would remove the word “aggressive”.

Fixed, now in line 309.

Line 275-277: This is speculative. It is not stated what dose of steroids the dog was receiving. For example, a dog with hypoadrenocorticism receives prednisone every day for years, but that dose is not immunosuppressive. An anti-inflammatory dose of prednisone is also not immunosuppressive. This sentence should be removed and the prior statement about the dog clarified (dose (mg/kg/day) if possible or amended.

The sentence has been removed, and the paragraph now reads: “The dog from which isolate AGFA24 was recovered, was an asymptomatic carrier dog that was a pack companion of a sick animal that died of an acute icteric syndrome of leptospirosis.” Lines 293 - 295

Line 283-303: The authors speculate about the reason this puppy had a reciprocal titer of 3200 to serovar Canicola and a reciprocal titer of 200 to serovar Icterohaemorrhagiae when the dog was infected with serovar Copenhageni. The authors proposed theory that the dog was co-infected with both serovars is illogical. Natural infection confers good cross-serovar protective immunity and the titer to serovar Canicola would suggest a staggered infection, not co-infection. Even then, establishment of a second infection would seem unlikely. It’s possible that the Canicola titer represents maternal antibody from vaccination or that the dog was previously vaccinated for serovar Canicola (although one would imagine the dog was vaccinated for Icterohaemorrhagiae, too).

Thank you, the paragraph was changed to this: “However, the high antibody titer observed against serovar Canicola in the absence of isolation of such a serovar remains obscure. One explanation might be the possibility of a cross reaction between both serovars, or the presence maternal derived antibodies [27].” Also, reference 27 was included to support this. Lines 338 – 340.

Line 301: What does “…no decrease in the urine density” mean? There is no context to understand this statement.

We fixed a little this paragraph, now in lines 335 -337. “In addition, it is possible that due to the severe acute infection, urinary density was within normal parameters, contrary to what is usually reported in subacute cases of canine leptospirosis [26].”

Line 321- (and abstract): This study did not compare the serum titer to Canicola for dogs not shedding leptospires, so stating that using the MAT has value in identifying carriers is misleading.

Certainly, in the first draft, it was not referred the result of MAT in the group of 58 asymptomatic dogs. A paragraph saying that has been inserted: “35 out of the 58 asymptomatic dogs (60.3%), showed antibody titers of at least 1:100 against Leptospira, 18 were males (51.4%) and 17 females (48.5%). The more frequently detected serovars by the MAT were Canicola (23, 39.6%), Bratislava (20, 34.4%), Pyrogenes (19, 32.7%), Grippotyphosa (11, 18.9%) and Icterohaemorrhagiae (7, 12%). 10 out of the 58 dogs (17%) (7 males and 3 females), showed antibody titers ≥ 1:3,200. Four Leptospira isolates (LOCaS28, 32, 34, and 46), were obtained out of the 58 asymptomatic sampled dogs (6.89%), all them were male dogs and all them showed a titer ≥ 1:3,200 against serovar Canicola” Lines 177-183

Likewise, I think it is inaccurate to characterize the dogs intended for destruction as “asymptomatic”. How long were these dogs held by the Canine Control Centre? What veterinary evaluation was performed? There was not any bloodwork (CBC, serum biochemistry profile) performed on these dogs, so stating that they were asymptomatic is potentially a gross mischaracterization.

When referring as “asymptomatic”, we mean that those animals showed no apparent sign of disease. That has been stated in lines 97-99 in this manner: “In an open study done to identify Leptospira carriers, 58 dogs with no apparent sign of disease (asymptomatic) and destined to destruction in a Canine Control Centre in Mexico City were sampled.”

The same can be said of the three “suspicious carrier dogs” from which only urine was collected. If not laboratory assessment of these dogs was performed, how can it be stated definitively that they were healthy dogs (or at least asymptomatic)?

Well, those were UADY22, CEL60 and AGFA24. In the first two, kidney samples were collected, because those two dogs came from similar epidemiological studies pursuing Leptospira isolation done in the past. For the AGFA24 isolate, we have explained the sampling conditions, and being that dog a asymptomatic animal which laboratory tests were performed, the only samples available were serum and urine. We have explained these two issues in lines 114 – 124: “Two additional Leptospira isolates obtained from the kidneys of two asymptomatic dogs were included in this study: CEL60 and UADY22 isolates from dogs included in two independent epidemiological studies. Those studies were done respectively in the city of Toluca, in central Mexico (Carmona-Gasca, personal communication), and in the city of Merida, Yucatan in southern Mexico (Cárdenas Marrufo and Vado Solís, personal communication). The isolate AGFA24 was obtained from another asymptomatic carrier dog; this individual was the companion of a dog with fatal icteric leptospirosis. A urine sample was taken after detection of a titer as high as 1:6,400 against serovar Canicola in the MAT. The urine sample from this suspicious carrier dog was obtained by cystocentesis practiced at the veterinary hospital and cultured directly in site in EMJH and Fletcher media.”

Line 326-: The authors cannot state that their dogs were “acting as carriers and disseminators of virulent leptospires”.  That potential may exist, but they don’t know that definitively.

We try to be careful in not stating that carriers are disseminating virulent leptospires for sure, but that this potential risk exist. Here the paragraph in lines 372-376: “High antibody titers against Leptospira have been usually associated with an active infection and disease or sometimes because of a recent vaccination [30]. However, our findings show that dogs acting as carriers and potential disseminators of virulent leptospires might reach high MAT titers against infecting Leptospira serovars.”

Line 331-: This is a poorly worded sentence, but also complete nonsense. For starters, this study has nothing to do with ehrlichiosis and it’s unclear why the authors now mention it.

That sentence has been deleted.

Do the authors suggest routine MAT testing of every dog every year?

Well, we want to say that it is important to detect potential asymptomatic dogs that may be infected with Leptospira and eliminate the organism through urine -as it was shown in the dog where AGFA24 isolate was recovered. So, veterinarians should be advised to think about potential risk factors for dogs to become Leptospira carriers and apply diagnostic tools to prove this. The MAT is a feasible test that can potentially detect that kind of carriers, as we have shown. In the 58 asymptomatic dog study, 10 dogs showed a titer of at least 1:3,200 (or higher), and four of those were shown as real carriers by isolation!! Even more, that some of those isolates were highly virulent. Applying that strategy was the way how we detected the AGFA24 carrier dog. That dog showed no sign of disease, the only thing that made us to suspect that it might be infected with Leptospira, was the risk factor of the companion dog that died of leptospirosis. So, we decided to take a serum sample, the MAT showed 1;6,400 against Canicola in that dog, we then took a urine sample by cystocentesis, and we were able to isolate a virulent serovar Canicola. In the same way, a veterinarian that considers a dog exposed to whatever risk factor, could use these criteria and might be able to detect potential Leptospira spreaders with a MAT, and not necessarily run a MAT testing on every single dog.

Since the authors didn’t do MAT testing on the 54 dogs that were destroyed that didn’t have leptospires identified in the kidney, they can’t make any statement as to the value of annual MAT testing. No other study to date would support that suggestion.

Well, I am sorry, and apologize for not having stated that in the first draft of our manuscript, but indeed we did!! We run MAT on each of the 58 dogs and cultured their kidneys and urine samples. Please accept my apologies. Now we have included a paragraph that explains that in lines 177-183: “35 out of the 58 asymptomatic dogs (60.3%), showed antibody titers of at least 1:100 against Leptospira, 18 were males (51.4%) and 17 females (48.5%). The more frequently detected serovars by the MAT were Canicola (23, 39.6%), Bratislava (20, 34.4%), Pyrogenes (19, 32.7%), Grippotyphosa (11, 18.9%) and Icterohaemorrhagiae (7, 12%). 10 out of the 58 dogs (17%) (7 males and 3 females), showed antibody titers ≥ 1:3,200. Four Leptospira isolates (LOCaS28, 32, 34, and 46), were obtained out of the 58 asymptomatic sampled dogs (6.89%), all them were male dogs and all them showed a titer ≥ 1:3,200 against serovar Canicola”

With kind regards.
